# Highly Enhanced Enzymatic Activity of Mn-Induced Carbon Dots and Their Application as Colorimetric Sensor Probes

**DOI:** 10.3390/nano11113046

**Published:** 2021-11-12

**Authors:** Ahyun Lee, Wooseok Kang, Jin-sil Choi

**Affiliations:** Department of Chemical and Biological Engineering, Hanbat National University, Daejeon 34158, Korea; leeah3238@gmail.com (A.L.); gne01041@gmail.com (W.K.)

**Keywords:** carbon dots, Mn ions, nanozyme, enzymatic activity, GABA detection, colorimetric sensor

## Abstract

Nanomaterial-based enzyme mimetics (nanozymes) have attracted significant interest because of their lower cost and higher stability compared to natural enzymes. In this study, we focused on improving the enzymatic properties of metal induced N-doped carbon dots (N-CDs), which are nanozymes of interest, and their applications for sensory systems. For this purpose, Mn(acetate)_2_ was introduced during the synthetic step of N-doped carbon dots, and its influence on the enzymatic properties of Mn-induced N-CDs (Mn:N-CDs) was investigated. Their chemical structure was analyzed through infrared spectroscopy and X-ray photoelectron spectrometry; the results suggest that Mn ions lead to the variation in the population of chemical bonding in Mn:N-CDs, whereas these ions were not incorporated into N-CD frameworks. This structural change improved the enzymatic properties of Mn:N-CDs with respect to those of N-CDs when the color change of a 3,3′,5,5′-tetramethylbenzidine/H_2_O_2_ solution was examined in the presence of Mn:N-CDs and N-CDs. Based on this enhanced enzymatic property, a simple colorimetric system with Mn:N-CDs was used for the detection of γ-aminobutyric acid, which is an indicator of brain-related disease. Therefore, we believe that Mn:N-CDs will be an excellent enzymatic probe for the colorimetric sensor system.

## 1. Introduction

In sensor systems, a chemical or physical transducer converts a chemical or biological signal into a recognizable signal (e.g., color or electrical signal). Natural enzymes are effective chemical transducers, and they are widely used in sensor systems [1,2]. For example, in the conventional enzyme-linked immunosorbent assay (ELISA), natural enzymes (e.g., horseradish peroxidase, HRP) catalyze different colorimetric reactions in the presence of substrates [3,4,5]. In addition, HRP accelerates H_2_O_2_-dependent one-electron oxidation, generating an electrical signal that allows for the sensitive detection of targets [6,7,8,9]. Regardless of their novel catalytic effectiveness, natural enzymes have limitations in industrial applications, such as low stability in harsh environmental conditions and relatively high costs of preparation, purification, and storage. Therefore, over the past few decades, researchers have made extensive efforts to develop replacements.

Nanomaterial-based enzyme mimetics (nanozymes) attract significant interest because of their low cost and high stability relative to those of natural enzymes, leading to their application in diverse fields including biosensing [10,11,12,13,14,15,16,17]. Among various nanozymes, carbon dots (CDs) are emerging materials owing to their unique properties [18,19,20,21]. Numerous researchers have investigated not only the enzymatic properties of CDs but also methods to improve these properties. For example, CDs that were doped with hetero ions such as S or P could easily catalyze 3,3′,5,5′-tetramethylbenzidine (TMB) to produce oxidized TMB in the presence of H_2_O_2_ [22,23]. The enzymatic properties of CDs are considerably improved through the formation of hybrids with other materials (e.g., Au, Pd, Fe_3_O_4_, Mn_3_O_4_, and CeO_2_) [24,25,26,27]. Metal ions (e.g., Mn and Fe) with multiple oxidation states promote the enzymatic activity of CDs [28,29,30,31]. The simple addition of metal ions and chelate molecules to the reaction solution accelerates the enzymatic activity of CDs at a neutral pH [32]. Doping metal species into a CD framework can act as a catalase mimic and reduce oxidative-stress-related damage on CDs by decomposing H_2_O_2_ to H_2_O and O_2_ [28]. Metal ions can also influence the formation of CDs and induce variations in the chemical structure of CDs during their synthesis, leading to the enhancement of physicochemical properties [29]. However, the enzymatic properties of metal-induced CDs have rarely been examined.

In this study, the effect of Mn on the formation of Mn-induced N-doped CDs (Mn:N-CDs) and the resulting enzyme-like properties of Mn:N-CDs were investigated. Mn(acetate)_2_ was added during the synthesis of N-doped CDs (N-CDs). However, Mn ions were not incorporated into the CD framework. The analysis of the chemical composition of Mn:N-CDs using X-ray photoelectron spectrometry (XPS) and infrared (IR) spectroscopy showed that Mn(acetate)_2_ leads to a change in the composition of the chemical bonding of Mn:N-CDs with respect to that of N-CDs synthesized without the addition of Mn(acetate)_2_. Mn:N-CDs show better peroxidase-like properties than N-CDs. Based on these enhanced enzymatic properties, Mn:N-CDs can be used as colorimetric sensor probes to detect various disease factors. In this study, γ-aminobutyric acid (GABA), which is known as a metabolite of the human microbiome and an indicator of various brain diseases, was successfully detected using a Mn:N-CD-based colorimetric sensor system.

## 2. Materials and Methods

### 2.1. Reagents

Citric acid (99.9%), ethylenediamine (>98%), Mn(acetate)_2_, H_2_O_2_, TMB (97%), GABA (≥99%), and a 3-(4,5-dimethylthiazol-2-yl)-2,5-diphenyltetrazolium bromide (MTT) assay kit were purchased from Sigma Aldrich (St. Louis, MO, USA) and used as received.

### 2.2. Equipment

The morphology and lattice distance of Mn:N-CDs were investigated via transmission electron microscopy (TEM; Tecnai G2 F30 S-Twin, FEI, Hillsboro, OR, USA) and the H-7650 system (Hitachi, Tokyo, Japan) installed at the Center for University-wide Research Facilities at Jeonbuk National University. X-ray diffraction (XRD) was performed using SmartLab (Rigaku, Tokyo, Japan). The chemical functional groups and composition of Mn:N-CDs were examined through Fourier-transform infrared (FT-IR) spectrometry (Nicolet 6700, ThermoFisher Scientific, Waltham, MA, USA) and XPS (K-Alpha+, ThermoFisher Scientific, Waltham, MA, USA), respectively. The amount of Mn in Mn:N-CDs was measured using inductively coupled plasma-mass spectrometry (ICP-MS; ELAN DRC II PerkinElmer, Waltham, MA, USA).

### 2.3. Synthesis of Mn:N-CDs

Mn:N-CDs were prepared by employing a hydrothermal method that is a modified synthetic method reported in our previous work [22]. In a typical synthesis procedure, citric acid (10 μmol), ethylenediamine (5 μmol), and Mn(acetate)_2_ (4.15 μmol) were dissolved in deionized (DI) water (15 mL). Then, the mixture was heated hydrothermally in a Teflon-equipped stainless-steel autoclave at 200 °C. After 1 h, the mixture was cooled to room temperature, and the residue was purified using column chromatography (CombiFlash NextGen 100, Teledyne ISCO, Lincoln, NE, USA) to obtain brown Mn:N-CDs.

### 2.4. Enzyme-Mimicking Activities of Mn:N-CDs

The peroxidase-like activity of Mn:N-CDs was measured using TMB as the substrate. Specifically, a 100 mM H_2_O_2_ solution, 10 mM TMB solution, and a Mn:N-CD suspension (500 μg·mL^−1^) were added to a citric acid buffer (150 mM, at pH 2), and the mixture was shaken thoroughly to ensure homogeneous mixing. The absorption of the sample at 652 nm was measured immediately using a multimode plate reader (SpectraMax M2e, Molecular Devices, LLC, San Jose, CA, USA). The relationship between the initial velocity, *V*, and the substrate concentration, [***S***], is given by the Michaelis–Menten equation as follows:(1)$$V=\frac{\left[S\right]V_{max}}{\left[S\right]+K_{m}}.$$

Here, *V_max_* is the maximum initial velocity of the enzymatic reaction, and *K_m_* is the Michaelis constant, which represents the concentration of the substrate at half the maximum velocity. The linear regression curve of the relationship between 1/*V* and 1/[*S*], viz. the Lineweaver–Burk plot, can be obtained by inverting the Michaelis–Menten equation to the following form:(2)$$\frac{1}{V}=\frac{K_{m}}{V_{max}}\frac{1}{\left[S\right]}+\frac{1}{V_{max}}.$$

In the plot of 1/*V* vs. 1/[*S*], the ordinate and abscissa intercepts represent the inverse of *V_max_* and −1/*K_m_*, respectively. *K_m_* and *V_max_* were estimated from these intercepts.

### 2.5. Detection of GABA Using the Colorimetric Sensor System

GABA was detected using TMB as the substrate, GABA (25–400 nM), and Mn:N-CD (0.5 mg·mL^−1^), all of which were mixed in a citric acid buffer (150 mM, pH 2) and incubated for 15 min. Then, H_2_O_2_ (100 mM) and TMB (10 mM) were added to the reaction solution, and the solution was vortexed for homogeneous mixing. Immediately after mixing, the absorption of the sample at 652 nm was measured at intervals of 1 min for 10 min using a multimode plate reader (SpectraMax M2e, Molecular Devices, LLC, San Jose, CA, USA).

## 3. Results

Mn:N-CDs were obtained using the previously reported hydrothermal method [22]. Citric acid and ethylenediamine were the sources of carbon, and their reaction molar ratio was 1:0.5. Mn(acetate)_2_ was also added, and the reaction ratio of Mn(acetate)_2_ to citric acid was 1:0.4 (Figure 1a). The TEM images shown in Figure 1a reveal that Mn:N-CDs possessed a round shape with an average diameter of 6.7 nm (Figure 1b). The high-resolution (HR) TEM image confirmed that Mn:N-CDs had a crystalline structure with a lattice distance of 0.24 nm, which matched the (100) plane of graphite (Figure 1c) [33,34]. In the XRD spectrum, Mn:N-CDs showed a broad peak at 19.14° corresponding to the (002) plane of graphite (JCPDS card no. 26-1076) (Appendix A) [35,36].

The chemical structure of Mn:N-CDs was further investigated using FT-IR and XPS. Comparing the FT-IR spectra of Mn:N-CDs and N-CDs indicated no differences in the composition of the chemical functional groups (Appendix A). However, the spectrum of Mn:N-CDs showed a significant increase in the intensity of the peak at 1560 cm^−1^, which corresponds to the stretching of the N–O bond [37,38,39]. Figure 2 shows the chemical composition of Mn:N-CDs obtained via XPS. In the XPS survey profile of Mn:N-CDs (Figure 2a), peaks originating from C_1s_, N_1s_, and O_1s_ can be clearly observed, whereas the Mn_2p_ peak is absent. The ICP-MS analysis (Appendix A) shows that the amount of Mn ions in Mn:N-CDs is negligible. Thus, Mn ions might not be doped or chelated in Mn:N-CDs. Based on the area of each peak in the XPS spectra (Appendix A), the atomic compositions of Mn:N-CDs and N-CDs are found to be similar. However, differences exist in the distributions of functional groups between Mn:N-CDs and N-CDs. The chemical bonding of each atomic component was investigated in detail through a spectral peak fitting of the high-resolution C_1s_, N_1s_, and O_1s_ spectra on the basis of the reported values of chemical bonds, as shown in Figure 2b–d and Appendix A. In the C_1s_ spectra (Figure 2b), a strong peak appears at 284 eV, corresponding to the C–C/C=C bond of the carbogenic domain, along with the peaks for C–O (285.5 eV) and C=O (288.0 eV). In the N_1s_ spectra (Figure 2c), Mn:N-CDs show a strong N_pyrrole_ peak (399.5 eV), which is found at the edge sites of the carbogenic domain, and weak N_graphite_ peak (401 eV), which is located in the sp^2^-carbon domain. Furthermore, the O_1s_ spectrum is composed of two peaks corresponding to C–O and C=O bonds (532.6 and 531.1 eV, respectively) (Figure 2d). Integrating the area of each fitted peak reveals that compared with N-CDs, Mn:N-CDs possess a higher proportion of C=O and N_pyroline_, which exist at the edge of carbogenic domains, and a lower proportion of C–C/C=C and N_graphene_ in carbogenic domains (Appendix A). The addition of Mn(acetate)_2_ appears to perturb the formation of the carbogenic domain, thereby increasing the amounts of functional groups on the edge of sp^2^ carbogenic domains.

Next, the enzyme-like activity of Mn:N-CDs is explored. In our previous study [22], N-CDs were confirmed to show peroxidase-mimicking activity; thus, the peroxidase-like activities of Mn:N-CDs are examined here by monitoring the change in the color of a mixed solution of TMB (10 mM) and H_2_O_2_ (100 mM) at pH 2 (Figure 3a). With the addition of Mn:N-CDs, the color of the TMB solution changes from pale yellow to blue owing to the oxidation of TMB (Figure 3b), whereas N-CDs cause a negligible change in the color of the TMB solution under the same experimental conditions. The time-dependent absorption at 652 nm is monitored in a steady-state kinetics study, and the result suggest that Mn:N-CDs exhibit significantly higher absorption at 652 nm than N-CDs, thereby presenting superior peroxidase-like activity (Figure 3c). The enzymatic activity of Mn:N-CDs is estimated by calculating the V_max_ and K_m_ values of the reaction in the presence of Mn:N-CDs using the Lineweaver–Burk equation (Figure 3d and Appendix A). Mn:N-CDs show a slightly lower V_max_ value (0.11 μM·s^−1^) than N-CDs (0.14 μM·s^−1^) (Figure 3d). This may be because compared with N-CDs, Mn:N-CDs have a lower amount of graphitic N atoms, which help generate radical oxygen species [40] (Appendix A). However, there is a large difference between the values of K_m_, which represents the affinity between a substrate and enzyme (Mn:N-CDs: 0.14 mM; N-CDs: 3.88 × 10^3^ mM) (Figure 3d). According to the values of K_m_, Mn:N-CDs show a higher affinity to H_2_O_2_ compared with N-CDs. This may be because the increased amount of hydrophilic functional groups facilitates the interaction between H_2_O_2_ and Mn:N-CDs, resulting in superior enzymatic activity compared with that of N-CDs. In addition, no noticeable toxicity is observed in Mn:N-CDs up to 0.5 mg·mL^−1^ when they are co-cultured with lung cancer cells (A549) and kidney cells (HEK293) for 24 h (Appendix A). Hence, Mn:N-CDs with high enzymatic activity and biocompatibility are suitable as a colorimetric probe in various biological applications.

As the color change originating from the enzyme-like activity of Mn:N-CDs can be easily observed with the naked eye, it is advantageous to apply them in colorimetric sensor systems. In recent years, numerous research groups have examined the metabolism of the human microbiome and its relevance to diseases [41,42,43,44], and information about the composition of the microbiome or its metabolites can be quite helpful for understanding their relevance [45]. GABA is not only a representative metabolite released in the metabolic process of the microbiome but also an inhibitory neurotransmitter that acts on the central nervous system of mammals [45,46,47]. Therefore, it can be an indicator of the association between intestinal microbiota and brain-related diseases. Because GABA contains a carboxyl group and an amine in one molecule, the net charge varies with the pH. At pH 2, where the peroxidase-like property of Mn:N-CDs is maximized (Appendix A), GABA can lead to the aggregation of anionic Mn:N-CDs through electrostatic attractions owing to the positive charge on GABA (Figure 4a). The aggregation of Mn:N-CDs can change their enzymatic activity owing to variations in their surface area or the pathway of electron migration. At pH 2, the ζ potentials of Mn:N-CDs and GABA are measured to be −1.56 and 1.74 eV, respectively, whereas the net charge of the mixture of Mn:N-CDs and GABA is evaluated as neutral (−0.17 eV). Unlike Mn:N-CDs only, when the mixture of GABA and Mn:N-CDs is added to the reaction solution containing TMB and H_2_O_2_, there is no recognizable change in the color of the solution (Figure 4b). When the absorbance (at 652 nm) of the reaction solution is monitored, the change in absorbance change is significantly suppressed by the addition of both GABA and Mn:N-CDs, in contrast with the reaction solution with Mn:N-CDs only (Figure 4c). On the other hand, the addition of other small molecules such citric acid, L-glutamic acid, and L-ascorbic acid does not perturb the enzymatic activity of Mn:N-CDs (Appendix A). GABA in artificial urine (BZ103, Biochemazone™, Edmonton, Canada) is also successfully detected using Mn:N-CDs (Appendix A). When the absorbance change of the solution is examined by adding GABA at different concentrations (0–400 nM), the absorbance decreases proportionally with the increasing GABA concentration. In particular, the absorbance change shows a linear correlation in the GABA concentration from 50 to 300 nM (Figure 4d and Appendix A). The limits of detection and quantitation are calculated as 59.191 nM and 179.366 nM, respectively (Appendix A).Considering that the ranges of GABA concentrations in the blood of normal people and Alzheimer patients are 100–200 nM [48,49], Mn:N-CD-based colorimetric sensor systems can be successfully applied for the indication of disease occurrence.

## 4. Conclusions

In this study, the nanozyme effect was remarkably increased using Mn(acetate)_2_. Although the exact role of Mn ions in the formation of Mn:N-CDs has not been explored fully, the addition of Mn(acetate)_2_ to the reaction solution seems to generate more functional groups at the edge of carbogenic domains in Mn:N-CDs than in N-CDs, resulting in improved peroxidase-like properties. Mn:N-CDs with strong enzymatic effects can be applied as a colorimetric sensor probe for the detection of GABA. Specifically, the simple addition of GABA significantly suppressed the enzymatic effect of Mn:N-CDs, which was observed using the variation in the intensity of the blue color of the solution. Therefore, Mn:N-CDs are promising candidates as sensing probes that can convert a biological signal to an easily recognizable color, and a Mn: N-CD-based detection system can be applied as a monitoring system in everyday life for the early diagnosis or maintenance of diseases. In addition, through combining with target-specific molecules, Mn:N-CDs can be further applied as sensitive and selective colorimetric sensor probes for the detection of various disease.

## Figures and Tables

**Figure 1 nanomaterials-11-03046-f001:**
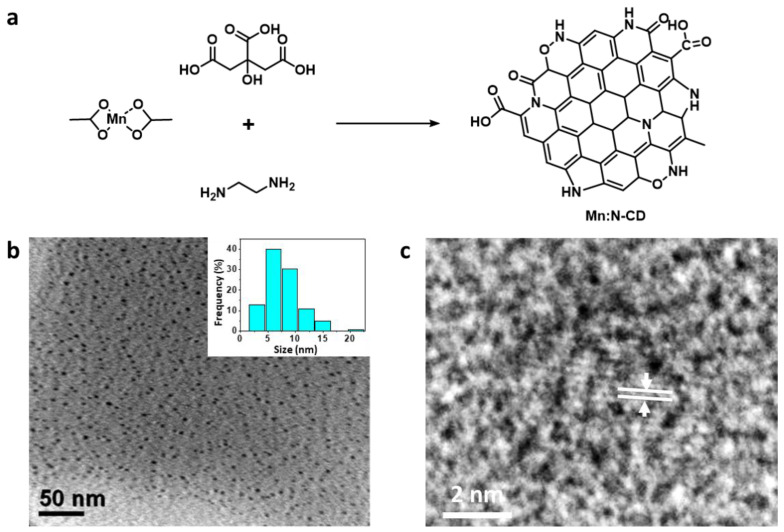
(**a**) Schematic of the synthesis of Mn-induced N-doped carbon dots (Mn:N-CDs). (**b**) Low-resolution and (**c**) high-resolution transmission electron microscopy images of the Mn:N-CDs. Inset in (**b**) particle size distribution.

**Figure 2 nanomaterials-11-03046-f002:**
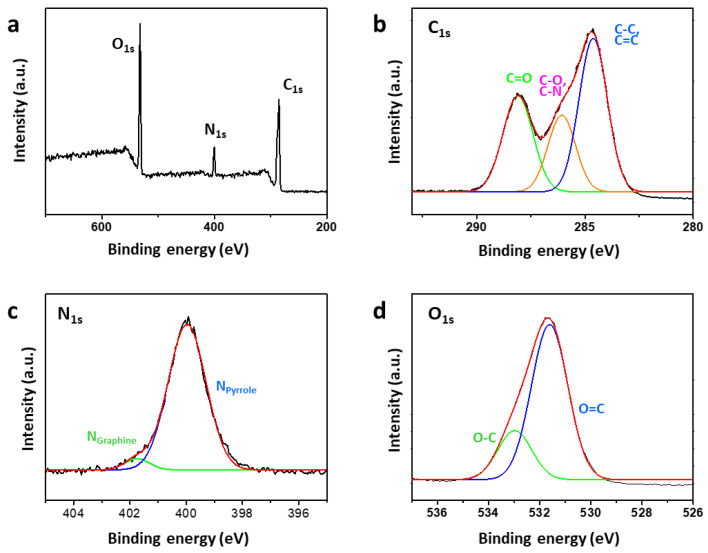
(**a**) X-ray photoelectron spectrometry survey scan profile and high-resolution (**b**) C_1s_, (**c**) N_1s_, and (**d**) O_1s_ spectra of Mn:N-CDs.

**Figure 3 nanomaterials-11-03046-f003:**
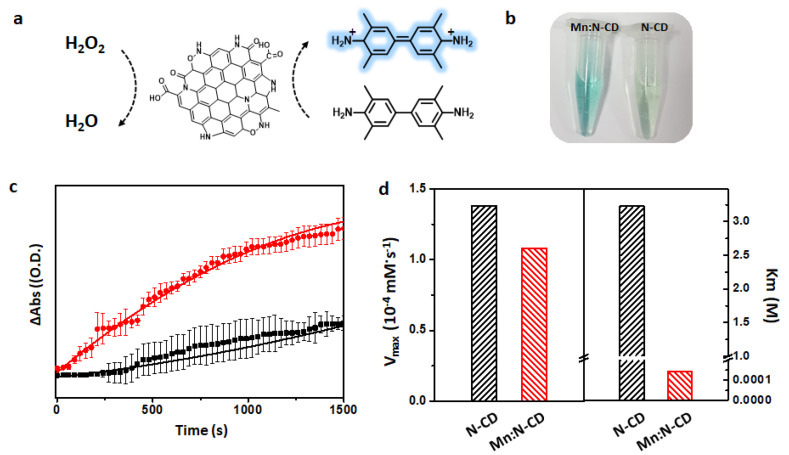
(**a**) Schematic of the peroxidase mimetic activity of Mn:N-CDs and (**b**) images of 3,3′,5,5′-tetramethylbenzidine (TMB)/H_2_O_2_ solutions with Mn:N-CDs or N-doped carbon dots (N-CDs). (**c**) Kinetic curves of Mn:N-CDs and N-CDs for monitoring the catalytic oxidation of TMB (10 mM phosphate buffer, pH = 2) with 100 mM H_2_O_2_ in the presence of the Mn:N-CDs or N-CDs. (**d**) Comparison of the *V*_max_ and *K*_m_ values of Mn:N-CDs and N-CDs; these were calculated from the Lineweaver–Burk plot (n = 3).

**Figure 4 nanomaterials-11-03046-f004:**
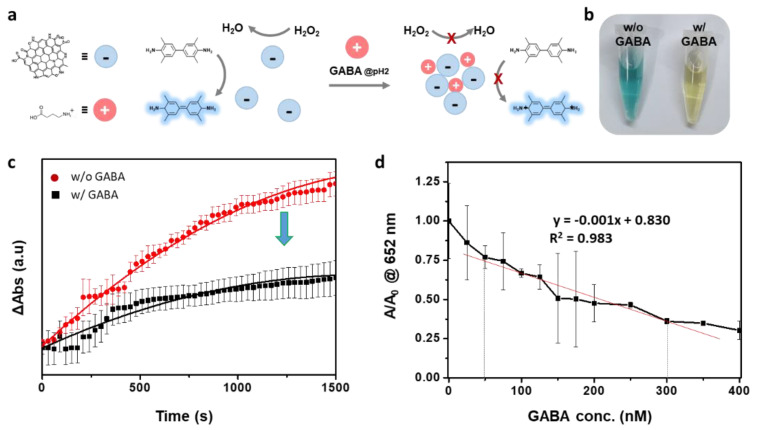
(**a**) Schematic of γ-aminobutyric acid (GABA) detection using Mn:N-CDs. (**b**) Images of TMB/H_2_O_2_/Mn:N-CDs solutions with or without GABA. (**c**) Kinetic curves of Mn:N-CDs with and without GABA for monitoring the catalytic oxidation of TMB (0.15 M citric acid buffer, pH = 2) with 100 mM H_2_O_2_. (**d**) Variation in absorption (at 652 nm) of the mixed solution containing Mn:N-CDs, GABA, TMB, and H_2_O_2_ as a function of the concentration of GABA.

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
