# Peer review of "Highly Enhanced Enzymatic Activity of Mn-Induced Carbon Dots and Their Application as Colorimetric Sensor Probes"

_nanomaterials, 2021, doi:10.3390/nano11113046_

Round 1

Reviewer 1 Report

In the present article, authors demonstrate an enhanced enzymatic activity on Mn-induced carbon dots for detection of γ-aminobutyric acid. To prove the concept, authors fabricated a Mn:N-CDs and well characterized. Further POD mimetic properties were examined and applied for the detection of GABA. Overall, present manuscript is having some significant advancements so suitable to be published in this journal; however, some points should be addressed before publication:

  1. Page 4, line 126, authors stated that “However, the spectrum of Mn:N-CDs shows a significant increase in the intensity of the peak at 1560 cm-1, which corresponds to the stretching of the N-O bond.” Cite reference.
  2. In page 5, POD mimetic experiments were carried out at pH=2. Recommended to check at biological pH (5/6/7).
  3. Page 5, line 174, authors mentioned “skin” cancer cells (A549), it’s a “lung” not “skin”. Change it.
  4. How about the cytotoxicity of Mn:N-CDs in normal cells?
  5. Page 6, line 209, authors mentioned that GABA concentrations in Alzheimer patients are 100–200 nM but authors performed their experiments between 0–400μM. Recommended to check with nM concentrations of GABA.
  6. Figure 4c, x-axis, is it nM  or μM?
  7. Figure 4b and Figure 4c legends missing in text.
  8. what is the lower and upper limit sensitivity of GABA on present nanozyme.
  9. How rich functional groups on Mn:N-CDs can improve the POD activity?
  10. Cite the following papers on POD mimetic nanozymes appropriately (https://doi.org/10.3390/biom11071015, https://doi.org/10.3390/catal10091009).

Author Response

Overall comment: In the present article, authors demonstrate an enhanced enzymatic activity on Mn-induced carbon dots for detection of γ-aminobutyric acid. To prove the concept, authors fabricated a Mn:N-CDs and well characterized. Further POD mimetic properties were examined and applied for the detection of GABA. Overall, present manuscript is having some significant advancements so suitable to be published in this journal; however, some points should be addressed before publication:

Our response: We appreciate the reviewer for their positive comments on our work. We have carefully considered the reviewer's comments, and these comments helped us to improve the quality of our work. We believe the revised manuscript now meets the high criteria for Nanomaterials.

Comment #1: Page 4, line 126, authors stated that “However, the spectrum of Mn:N-CDs shows a significant increase in the intensity of the peak at 1560 cm-1, which corresponds to the stretching of the N-O bond.” Cite reference.

Our response: We have added three references (#38–40 in the revised manuscript) in which the IR peak of anisotropic N–O vibration was reported at ca. 1560 cm-1.

# 38: J. Am. Chem. Soc. 1956, 78, 4225–4229.

# 39: Bulgarian Chemical Communications 2013, 45, 24–31.

# 40: Recueil des Travaux Chimiques des Pays-Bas 1957, 76, 801–809.

Comment #2: In page 5, POD mimetic experiments were carried out at pH=2. Recommended to check at biological pH (5/6/7).

Our response: Following the reviewer’s comment, the enzyme activity of Mn:N-CDs at various pH conditions (pH 2–9) was examined. As shown in Figure S6 in the revised Supporting Information, Mn:N-CDs show their maximum enzymatic activity at pH 2 and 3, whereas their enzymatic activity is very low at the pH 4 and above.

Comment #3: Page 5, line 174, authors mentioned “skin” cancer cells (A549), it’s a “lung” not “skin”. Change it.

Our response: We appreciate the reviewer’s correction. We have corrected the type of cancer cells, A549.

Comment #4: How about the cytotoxicity of Mn:N-CDs in normal cells?

Our response: We tested the toxicity of Mn:N-CDs on kidney cells, HEK293, using MTT assay. Similar to the case of A549, Mn:N-CDs exhibit negligible toxicity in HEK293. The toxicity results have been added in the Supporting Information, Figure S5.

Comment #5: Page 6, line 209, authors mentioned that GABA concentrations in Alzheimer patients are 100–200 nM but authors performed their experiments between 0–400μM. Recommended to check with nM concentrations of GABA. Figure 4c, x-axis, is it nM  or μM?

Our response: We thank the reviewer for the important comment on our error. In fact, the GABA concentration range is 0–400 nM, not 0–400 μM, as described in the experimental section and Figure 4c. We have corrected the information on GABA concentration in the revised manuscript.

Comment #6: Figure 4b and Figure 4c legends missing in text.

Our response: We have added legends for Figure 4b and Figure 4c in the revised manuscript on page 7, lines 230 and 233, respectively

Comment #7: what is the lower and upper limit sensitivity of GABA on present nanozyme.

Our response: Following the reviewer’s comment, the detection limit of GABA using Mn:N-CDs has been investigated. When the absorption at 652 nm of the reaction solution containing TMB, H2O2, and Mn:N-CDs was monitored with the addition of GABA concentration between 0 and 1,000 nM, the lower and upper detection limits of GABA using Mn:N-CDs are determined to be 50 and 400 nM, respectively. The result was newly added to the revised Supporting Information as Figure S8.

Comment #8: How rich functional groups on Mn:N-CDs can improve the POD activity?

Our response: Rich functional groups (e.g., COOH, NO2, OH) on Mn:N-CDs can increase the hydrophilicity of sp2 domains in CD with respect to those on N-CDs. Some studies have shown that the interaction of CDs with molecules or targets can be regulated by modulating the functional groups of the CDs (J .Colloid Interf. Sci. 2019, 554, 722–730; Adv. Phys.: X 2020, 5, 1758592). Similarly, we believe the rich hydrophilic functional groups in Mn:N-CDs might increase the interactions of Mn:N-CDs with hydrophilic H2O2 and TMB molecules. 

Comment #9: Cite the following papers on POD mimetic nanozymes appropriately

(https://doi.org/10.3390/biom11071015, https://doi.org/10.3390/catal10091009).

Our response: We have cited the recommended papers as references #16 and #17 in the revised manuscript.

Reviewer 2 Report

the manuscript entitled "Highly enhanced enzymatic activity of Mn-induced carbon dots for detection of γ-aminobutyric acid" describes a novel protocol for CDs production with enhanced enzymatic features. The production and characterization of these improved CDs are consistent and well presented, but the application for GABA detection requires many improvements. First, if the GABA concentration in blood is present in nanomolar concentration why this sensor could be useful if it is able to provide responses in micromolar concentration ranges? Also, where is the specificity or at least the selectivity of the sensor? Is this sensor capable to detect GABA and discriminate it among all the analytes present in the blood? The authors do not present a detection limit, an equation of the linear response with linked R2, an interference study, a recovery study, a matrix effect. Did the author perform the analysis in blood? From an analytical point of view, the manuscript is very poor.

Author Response

Overall comment: the manuscript entitled "Highly enhanced enzymatic activity of Mn-induced carbon dots for detection of γ-aminobutyric acid" describes a novel protocol for CDs production with enhanced enzymatic features. The production and characterization of these improved CDs are consistent and well presented, but the application for GABA detection requires many improvements.

Our response: First, we appreciate the reviewer for their positive comments on the characterization of CDs in our work. In addition, we agree that some improvement in the GABA detection is required. Therefore, we have carefully considered the reviewer's comments, and these comments helped us to improve the quality of our work.

Comment #1: First, if the GABA concentration in blood is present in nanomolar concentration why this sensor could be useful if it is able to provide responses in micromolar concentration ranges?

Our response: We appreciate the reviewer’s question. As mentioned in the answer to reviewer #1’s comment #5, the GABA concentration that we tested in this study ranged from 0 to 400 nM; thus, we mentioned that Mn:N-CDs can be useful for the detection of GABA in biological fluids. We have corrected the information on the GABA concentration in the revised manuscript.  

Comment #2: Is this sensor capable to detect GABA and discriminate it among all the analytes present in the blood? Did the author perform the analysis in blood?

Our response: We appreciate the reviewer’s valuable questions. In fact, GABA was not detected in blood in this study. Because the Mn:N-CD-based sensor is a colorimetric sensor, the analytes must be purified. In addition, other components in biological fluids can easily interfere with the electrostatic interactions between Mn:N-CDs and the target However, we believe the target specificity in the biological fluid can be improved by conjugation with specific targeting agents, which can be developed in future studies.

Comment #3: The authors do not present a detection limit, an equation of the linear response with linked R2, an interference study, a recovery study, a matrix effect. Also, where is the specificity or at least the selectivity of the sensor? From an analytical point of view, the manuscript is very poor.

Our response: Following the reviewer’s comment, additional experiments have been performed to validate the applicability of Mn:N-CDs as colorimetric sensor probes.

  1. Detection limits: As mentioned in the response to reviewer #1’s comment #7, the lower and upper detection limits of GABA using Mn:N-CDs are determined to be 50 and 400 nM, respectively. The information has been newly added in the revised Supporting Information as Figure S8.
  2. Equation of the linear response with linked R2: The equation of the linear response in the GABA concentration range between 50 to 300 nM was determined to be y = 0.001x + 0.830 (R2 = 0.983). The equation and R2 value are added to the revised Figure 4c.
  3. Specificity: We compared the detection capability of Mn:N-CDs on citric acid, L-glutamic acid, and L-aspartic acid. Unlike GABA, the addition of three molecules mentioned above to the reaction solution does not result in a color change owing to the weak interactions between the negatively charged molecules and Mn:N-CDs at pH 2. The result has been newly added in the revised Supporting Information as Figure S6.

In fact, we mainly focused on improving the enzymatic effect of Mn:N-CDs, and GABA was detected as a case study to show their applicability as sensor probes. Thus, our Mn:N-CD-based GABA detection system possesses limitations, including selectivity/specificity in biological fluid and the recovery of targets. Therefore, in future studies on the development of sensor systems using Mn:N-CDs, the factors that the reviewer mentioned will be considered seriously. To avoid confusion, the title “Highly enhanced enzymatic activity of Mn-induced carbon dots for detection of γ-aminobutyric acid” has been changed to “Highly enhanced enzymatic activity of Mn-induced carbon dots and their application as colorimetric sensor probes.”

Round 2

Reviewer 2 Report

The analytical data presented by the authors were and remain poor. De facto, the biosensor presented was not tested in real samples, any interferent was tested, the detection limit was not calculated (the lower and upper limits of 50 and 200 nM represents the limits of a linear range of concentrations were the biosensor respond, but the author should calculate the Limit of Detection (LOD) according to the available equations (e.g. LOD = 3.3(Sy/S)). There is a huge lack in the analytical features of the biosensor, even if the nanomaterials exploited demonstrated to enhance the enzymatic performance; for this reason, the manuscript is not suitable for this Journal. I suggest to re-think the article better focusing on the nanomaterial and its improvemets.

Author Response

The analytical data presented by the authors were and remain poor. De facto, the biosensor presented was not tested in real samples, any interferent was tested.

Our response: The GABA detection capability of Mn:N-CDs in artificial urine (BZ103, Biochemazone™, USA) has been evaluated. To imitate the real biological sample, GABA is added to artificial urine which possesses similar composition and ingredients (biological contents, mineral composition, enzymes, and pH) of natural urine. As shown in Figure S8 in the revised supporting information, the enzymatic activity of Mn:N-CDs was suppressed even with the addition of GABA containing artificial urine.

Comment #2: The detection limit was not calculated (the lower and upper limits of 50 and 200 nM represents the limits of a linear range of concentrations were the biosensor respond, but the author should calculate the Limit of Detection (LOD) according to the available equations (e.g. LOD = 3.3(Sy/S)).

Our response: LOD was calculated as 59.191 nM on average and the information of LOD has been newly added as Figure S10 in the revised supporting information.

Comment #3: There is a huge lack in the analytical features of the biosensor, even if the nanomaterials exploited demonstrated to enhance the enzymatic performance; for this reason, the manuscript is not suitable for this Journal. I suggest to re-think the article better focusing on the nanomaterial and its improvemets.

Our response: Although we agree that our probe system needs to be improved to be applied for the real biological sample, we still believe our study provides significance in the study to develop a sensor probe for GABA detection.

The current detection system of GABA is very limited. Most of the studies on GABA detection are done using HPLC or ELISA systems which need a purification process and expensive equipment. Recent studies provide electrode-based GABA sensors with high sensitivity (Anal. Chem. 2018, 90, 3067-3072; Nano Convergence 2019, 6, Article number: 13). However, those systems still need to verify their detection capability in biological samples and the cost for the production of the sensor is quite expensive. 

On the other hand, compared to other sensor systems, our Mn:N-CD-based sensory system possess advantages as described below:

  1. recognizable with the naked eye.
  2. simple to perform
  3. capable to work with a biological sample
  4. low cost for the production
  5. high sensitivity (nM scale detection)

Therefore, our Mn:N-CDs, which exhibit highly enhance enzymatic properties, can provide a possibility to build a new type of GABA detection system which can redeem the existing systems. Therefore, we believe our study can contribute to “Nanomaterials” which aims for “any application of new nanomaterials or new application of nanomaterials”.
